# Optimality of variational inference for stochastic block model with missing links

**Solenne Gaucher**
Département de Mathématiques d'Orsay, Université Paris-Saclay, Orsay, France
`solenne.gaucher@math.u-psud.fr`

**Olga Klopp**
ESSEC Business School, Cergy, France
CREST, ENSAE, Palaiseau, France
`kloppolga@math.cnrs.fr`

## Abstract

Variational methods are extremely popular in the analysis of network data. Statistical guarantees obtained for these methods typically provide asymptotic normality for the problem of estimation of global model parameters under the stochastic block model. In the present work, we consider the case of networks with missing links that is important in application and show that the variational approximation to the maximum likelihood estimator converges at the minimax rate. This provides the first minimax optimal and tractable estimator for the problem of parameter estimation for the stochastic block model with missing links. We complement our results with numerical studies of simulated and real networks, which confirm the advantages of this estimator over current methods.

## 1 Introduction

The analysis of network data poses both computational and theoretical challenges. Most results obtained in the literature concentrate on the stochastic block model (SBM) which is known to be a good proxy for more general models, such as the inhomogeneous random graph model, [35]. Recently, variational methods ([28, 49]) have attracted considerable attention as they offer computationally tractable algorithms often combined with theoretical guarantees. Theoretical results that one can find for such variational methods provide asymptotic normality rates for parameter estimates of stochastic block data. For example, consistency has been shown for profile likelihood maximization [8] and variational approximation to the maximum likelihood estimator [13], [7]. These results have been extended to the case of dynamic stochastic block model [34] and sampled data [48]. These work focus on parameter estimation, as in [44] and [53], who establish the minimax optimality of variational methods in a large class of models (which does however not include the stochastic block model). Variational inference has also been successfully applied to the problem of community detection, see, e.g., [3, 54, 26, 45]. In particular, the authors of [54] show that an iterative Batch Coordinate Ascent Variational Inference algorithm designed for the two-parameters, assortative stochastic block model achieves statistical optimality for community detection problem. Note that this algorithm cannot be extended to the more general stochastic block model considered here.

In parallel with this line of work, the problem of statistical estimation of model parameters, in particular, the question of minimax optimal convergence rates, has been actively studied in the statistical community. In the case of dense graphs, a pioneering paper [17] shows that, for the problem of estimating the matrix of connection probabilities, the least square estimator is minimax optimal and [18] provides optimal rate for Bayes estimation. For the more challenging case of

35th Conference on Neural Information Processing Systems (NeurIPS 2021).

sparse graphs, the minimax optimal rates have been first obtained in [29] building on the restricted least square estimator. In [16], the authors consider the least square estimator in the setting when observations about the presence or absence of an edge are missing independently at random with the same probability $p$. Unfortunately, least square estimation is too computationally expensive to be used in practice. Many other approaches have been proposed, for example, spectral clustering [39, 22, 46], modularity maximization [42, 8], belief propagation [14], neighbourhood smoothing [55], convex relaxation of k-means clustering [20] and of likelihood maximization [4], and universal singular value thresholding [11, 30, 51]. These approaches are computationally tractable but show sub-optimal statistical performances. So the question of possible computational gap when no polynomial time algorithm can achieve minimax optimal rate of convergence has been raised.

The present work goes in these two directions. We study the statistical properties of the mean field variational Bayes method and show that it achieves the optimal statistical accuracy. In particular, these results close the open question on the possible existence of a computational gap for the problem of global parameter estimation. We built our analysis on the approach developed in [13], [7] and [48] using the closeness of maximum likelihood and maximum variational likelihood and on the results that show the minimax optimality of the maximum likelihood estimator [19].

In the present paper, we deal with settings where the network is not fully observed, a common problem when studying real life networks. In many applications the network has missing data as detecting interactions can require significant experimental effort, see, [32, 52, 24, 21]. For example, in biology graphs are used to model interactions between proteins. Discovery of these interactions can be costly and time-consuming [9]. On the other hand, the size of some networks from social media or genome sequencing may be so large that only subsamples of the data are considered [6]. It has been observed that incomplete observation of the network structure may considerably affect the accuracy of inference methods [31] and missing data must be taken into account while analyzing networks data. A popular approach consists in considering the edges with uncertain status as non-existing. In the present paper, we use a different framework by considering such edges as missing and introducing a separated data missing mechanism. A natural application of our method is link prediction [36, 56], the task of predicting whether two nodes in a network are connected. Our approach allows to deduce the pairs of nodes that are most likely to interact based on the known interactions in the network. Behind inference of the networks structure, our algorithms can be used to predict the links that may appear in the future if we consider networks evolving over the time. For example, in a social network, two users that are not yet connected but are likely to be connected can be recommended as promising friends.

## 1.1 Contribution and outline

The paper is organized as follows. After summarizing notations, we introduce our model and the maximum likelihood estimator for the stochastic block model with missing observations in Section 2. In Section 3, we introduce the mean field variational Bayes method and present a new estimator which combines the labels obtained using the variational method and the empirical mean for estimation of connection probabilities. In Section 3.2, we show that our estimator is minimax optimal for dense stochastic block models with missing observations as well as for sparse stochastic block models. Finally, in Section 4 we provide an extensive numerical study both on synthetic and real-life data which shows clear advantages of our estimator over current methods.

## 1.2 Notations

We provide here a summary of the notations used throughout the paper. For all $d \in \mathbb{N}_*$, we denote by $[d]$ the set $\{1, ..., d\}$. For $z : [k] \to [n]$ and all $(a, b) \in [d] \times [d]$, we abuse notations and denote $z^{-1}(a, b) = \{(i, j) : z(i) = a, z(j) = b, i \neq j\}$. For any two label functions $z, z'$, we write $z \sim z'$ if there exists a permutation $\sigma$ of $\{1, ..., k\}$ such that $(z(\sigma(a)))_{a \leq k} = (z(a))_{a \leq k}$. For any set $\mathcal{S}$, we denote by $|\mathcal{S}|$ its cardinality. For any matrix $\boldsymbol{A}$, we denote by $\boldsymbol{A}_{ij}$ its entry on row $i$ and column $j$. If $\boldsymbol{A} \in [0, 1]^{n \times n}$ and $\boldsymbol{A}$ is symmetric, we write $\boldsymbol{A} \in [0, 1]^{n \times n}_{\text{sym}}$. We denote by $\boldsymbol{A} \odot \boldsymbol{B}$ the Hadamard product of two matrices $\boldsymbol{A}$ and $\boldsymbol{B}$. The Frobenius norm of a matrix $\boldsymbol{A}$ is denoted by $\|\boldsymbol{A}\|_2 = \sqrt{\sum_{i,j} A_{ij}^2}$. We denote by $C$ and $C'$ positive constants that can vary from line to line. These are absolute constants unless otherwise mentioned. For any two positive sequences $(a_n)_{n \in \mathbb{N}}$, $(b_n)_{n \in \mathbb{N}}$, we write $a_n = \omega(b_n)$ if $a_n/b_n \to \infty$.

## 2  Maximum likelihood estimation in the stochastic block model with missing links

### 2.1  Network model and missing data scheme

In the simplest situation, a network can be represented as undirected, unweighted graph with $n$ nodes indexed from 1 to $n$. Then, the network can be encoded by its *adjacency matrix* $\boldsymbol{A} = (A_{ij})$. The adjacency matrix is a $n \times n$ symmetric matrix such that for any $i < j$, $\boldsymbol{A}_{ij} = 1$ if there exists an edge between node $i$ and node $j$, $\boldsymbol{A}_{ij} = 0$ otherwise. We consider that there is no edge linking a node to itself, so $\boldsymbol{A}_{ii} = 0$ for any $i$. A common approach in network data analysis is to assume that the observations are random variables drawn from a probability distribution over the space of adjacency matrices. More precisely, for $i < j$ the variables $\boldsymbol{A}_{ij}$ are assumed to be independent Bernoulli random variables of parameter $\boldsymbol{\Theta}^*_{ij}$, where $\boldsymbol{\Theta}^* = (\boldsymbol{\Theta}^*_{ij})_{1 \le i < j \le n}$ is a $n \times n$ symmetric matrix with zero diagonal entries. The matrix $\boldsymbol{\Theta}^*$ corresponds to the matrix of probabilities of observing an edge between nodes $i$ and $j$. This model is known as the *inhomogeneous random graph* model:

$$\forall 1 \le i < j \le n, \ \boldsymbol{A}_{ij}|\boldsymbol{\Theta}^*_{ij} \overset{ind.}{\sim} \text{Bernoulli}\left(\boldsymbol{\Theta}^*_{ij}\right). \tag{1}$$

Our focus is on the problem of estimation of the generative matrix $\boldsymbol{\Theta}^*$ which determines the overall structure of the network. This question is of particular interest for the task of link prediction.

Many of real-life networks are characterized by block structure. Loosely speaking, the block structure means that the nodes of the network are partitioned into groups called blocks, and that the distribution of the connections between nodes depends on the blocks to which the nodes belong. For example, when considering citation networks, where two articles are linked if one is cited by the other, it amounts to saying that the probability that two articles are linked only depends on their topic. Similarly, if one considers students of a school in a social network, it is a reasonable assumption to say that the probability that two students are linked only depends on their cohorts.

A very popular model that formalizes this idea is the stochastic block model (see, e.g., [27]). In this model, nodes are classified into $k$ communities: each node $i$ is associated with a community $z^*(i)$, where $z^* : [n] \to [k]$ is called the label function. This label function can either be treated as a parameter to estimate, or as a latent variable. In this last case, it is assumed that the indexes follow a multinomial distribution: $\forall i, z^*(i) \overset{i.i.d}{\sim} \text{Multinomial}(1; \alpha^*)$ where $\forall a \in [k]$, $\alpha_a$ is the probability that node $i$ belongs to the community $a$. Given this label function, the probability that there exists an edge between nodes $i$ and $j$ depends only on the communities of $i$ and $j$. Thus, the matrix of connection probabilities $\boldsymbol{\Theta}^*$ can be factorized as follows: $\boldsymbol{\Theta}^*_{ij} = \boldsymbol{Q}^*_{z^*(i)z^*(j)}$, with $\boldsymbol{Q}^*$ a $k \times k$ symmetric matrix such that $\boldsymbol{Q}^*_{ab}$ is the probability that there exists an edge between a given member of the community $a$ and a given member of the community $b$. The conditional stochastic block model can be written as:

$$\exists \boldsymbol{Q}^* \in [0,1]^{k \times k}_{\text{sym}}, \exists z^* : [n] \to [k]$$
$$\forall 1 \le i < j \le n, \ \boldsymbol{A}_{ij}|(\boldsymbol{Q}^*, z^*) \overset{ind.}{\sim} \text{Bernoulli}\left(\boldsymbol{Q}^*_{z^*(i)z^*(j)}\right), \ \boldsymbol{A}_{ii} = 0. \tag{2}$$

Assuming that the network follows the stochastic block model, the problem of estimating the matrix of connection probabilities reduces to estimating the label function $z^*$ and the matrix of probabilities of connections between communities $\boldsymbol{Q}^*$. Note that the conditional stochastic block model is at best identifiable up to a simultaneous permutation of the communities and of the rows and columns of the parameters $\boldsymbol{Q}^*$.

The stochastic block model has attracted considerable interest from the learning community. An important line of work has focused on the problem of estimation of the latent variables $z^*$, see, for example, [38, 10, 1, 41]. The best understood framework is the binary, balanced, symmetric, assortative block model. In this simpler model, the two communities have the same size, the same probability of intra-community connection ($\boldsymbol{Q}^*_{11} = \boldsymbol{Q}^*_{22} = p$), and nodes are assumed to be more connected with nodes of the same community ($p > q = \boldsymbol{Q}^*_{12}$). Much work has been done on the precise characterisation of the conditions on $p, q$ that allow for strong recovery of $z^*$, i.e. to estimate $z^*$ exactly with high probability. Closest to model (2) is perhaps the setting considered in [15]. In this work, the authors consider the related problem of community recovery in the binary block model [23],[2], and provide tight bounds on the recovery threshold for the balanced, two communities stochastic block model with missing observations. They propose a computationally

efficient algorithm for estimating $z^*$ in regime where strong recovery is possible; this, however, requires prior knowledge of the parameter $\boldsymbol{Q}^*$.

**Missing observations scheme** Usually, when working with network data, not all the edges are observed. To account for this situation we introduce $\boldsymbol{X} \in \{0,1\}_{sym}^{n \times n}$ the known sampling matrix where $\boldsymbol{X}_{ij} = 1$ if $\boldsymbol{A}_{ij}$ is observed and $\boldsymbol{X}_{ij} = 0$ otherwise. We assume that $\boldsymbol{X}$ is random and independent from the adjacency matrix $\boldsymbol{A}$ and its expectation $\boldsymbol{\Theta}^*$. For any $1 \leq i < j \leq n$, its entries $\boldsymbol{X}_{ij}$ are mutually independent and $\boldsymbol{X}_{ij} \stackrel{ind.}{\sim} \text{Bernoulli}(p)$ for some sampling rate $p \to 0$ such that $p = \omega\left(\log(n)/n\right)$ when $n \to \infty$.

## 2.2 Conditional maximum likelihood estimator

The log-likelihood of the parameters $(z, \boldsymbol{Q})$ with respect to the adjacency matrix $\boldsymbol{A}$ and the sampling matrix $\boldsymbol{X}$ is given by

$$\mathcal{L}_{\boldsymbol{X}}(\boldsymbol{A}; z, \boldsymbol{Q}) = \sum_{1 \leq i < j \leq n} \boldsymbol{X}_{ij}\left(\boldsymbol{A}_{ij}\log(\boldsymbol{Q}_{z(i)z(j)}) + (1 - \boldsymbol{A}_{ij})\log(1 - \boldsymbol{Q}_{z(i)z(j)})\right)$$

$$= \sum_{a \leq b} \log(\boldsymbol{Q}_{ab}) \sum_{(i,j) \in z^{-1}(a,b)} \boldsymbol{X}_{ij}\boldsymbol{A}_{ij} + \sum_{a \leq b} \log(1 - \boldsymbol{Q}_{ab}) \sum_{(i,j) \in z^{-1}(a,b)} \boldsymbol{X}_{ij}(1 - \boldsymbol{A}_{ij}).$$

Let us denote by $\mathscr{Z}_{n,k}$ the set of all label functions $z : [n] \to [k]$. For a given label function $z \in \mathscr{Z}_{n,k}$, the log-likelihood is maximized by taking

$$\boldsymbol{Q}_{ab} = \frac{\sum_{(i,j) \in z^{-1}(a,b)} \boldsymbol{X}_{ij}\boldsymbol{A}_{ij}}{\sum_{(i,j) \in z^{-1}(a,b)} \boldsymbol{X}_{ij}}.$$

It is interesting to note that, for a fixed label function $z$, maximizing the likelihood or minimizing the least square criterion defined as $\mathcal{C}_{\boldsymbol{X}}(\boldsymbol{A}; z, \boldsymbol{Q}) = \sum_{i<j} \boldsymbol{X}_{ij}\left(\boldsymbol{A}_{ij} - \boldsymbol{Q}_{z(i),z(j)}\right)^2$ yields the same estimator for the matrix $\boldsymbol{Q}$. The main difference between these two methods is rooted in the label functions selected by the two criteria, see, e.g. [19].

To bound the risk of the maximum likelihood estimator, it is usual to assume that there exists sequences $\rho_n$ and $\gamma_n$ such that $\forall i < j$,

$$0 < \gamma_n \leq \boldsymbol{\Theta}_{ij}^* \leq \rho_n < 1. \tag{3}$$

This assumption ensures that the loss associated to the maximum likelihood estimator is Lipschitz continuous. See, for example, [7] and [50], where the authors assume that the adjacency matrix is generated by an homogeneous stochastic block model for which the matrix $\boldsymbol{Q}^*/\rho_n$ has entries bounded away from $0$.

The restricted maximum likelihood estimator, $\widehat{\boldsymbol{\Theta}}$, is based on the maximization of the likelihood among block constant matrices with entries in $[\gamma_n, \rho_n]$:

$$\widehat{\boldsymbol{\Theta}}_{i<j} = \widehat{\boldsymbol{Q}}_{\widehat{z}(i)\widehat{z}(j)}, \ \widehat{\boldsymbol{\Theta}}_{ii} = 0$$
$$(\widehat{\boldsymbol{Q}}, \widehat{z}) \in \underset{\boldsymbol{Q} \in [\gamma_n, \rho_n]_{sym}^{k \times k}, z \in \mathscr{Z}_{n,k}}{\arg\max} \mathcal{L}_{\boldsymbol{X}}(\boldsymbol{A}; z, \boldsymbol{Q}). \tag{4}$$

In (4), $\gamma_n$ and $\rho_n$ are assumed to be known (see [19] for a discussion on how to estimate these parameters). Note that the Expectation-Maximization algorithm used in practice to obtain the variational approximation to the maximum likelihood estimator does not require the knowledge of these parameters. We also assume that $k$ is known and that it can depend on the number of nodes $n$; it can be chosen using a network cross-validation method [12], a sequential goodness-of-fit testing procedure [33] or a likelihood-based model selection method [50]. The following result provides the upper bound on the estimation risk of the maximum likelihood estimator:

**Theorem 1** (Corollary 2 in [19])**.** *Assume that $\boldsymbol{A}$ is drawn according to the conditional stochastic block model and $\rho_n = \omega(n^{-1})$. Then, there exists absolute constants $C, C' > 0$ such that, with probability at least $1 - 9\exp\left(-C\rho_n\left(k^2 + n\log(k)\right)\right)$,*

$$\|\boldsymbol{\Theta}^* - \widehat{\boldsymbol{\Theta}}\|_2^2 \leq C'\left(\frac{\rho_n^2}{((1-\rho_n)^2 \wedge \gamma_n^2)}\right)\frac{\rho_n\left(k^2 + n\log(k)\right)}{p}. \tag{5}$$

When all network entries are observed, we have $p = 1$. Note that this results implies that, when $\rho_n = O(\gamma_n)$, the maximum likelihood estimator is minimax optimal (see, [29, 16] for a statement of the lower bound).

## 3 Variational approximation to the maximum likelihood estimator

### 3.1 Definition of the estimator

The optimization of the log-likelihood function $\mathcal{L}_{\boldsymbol{X}}$ requires a search over the set of $k^n$ labels. As a consequence, the maximum likelihood estimator defined in (4) is computationally intractable. Celisse et al. [13] and Bickel et al. [7] are the first to study a variational approximation to this estimator. More recently, the authors of [48] used variational methods to approximate the maximum likelihood estimator in networks with missing observations. We start by formally introducing the variational approximation to the maximum likelihood estimator. We consider a stochastic block model with random labels with parameters $(\alpha, \boldsymbol{Q})$. For this model, the likelihood of the observed adjacency matrix $\boldsymbol{A}$ and sampling matrix $\boldsymbol{X}$ is given by

$$l_{\boldsymbol{X}}(\boldsymbol{A}; \alpha, \boldsymbol{Q}) = \sum_{z \in \mathscr{Z}_{n,k}} \left( \prod_{i \leq n} \alpha_{z(i)} \right) \exp\left( \mathcal{L}_{\boldsymbol{X}}(\boldsymbol{A}; z, \boldsymbol{Q}) \right).$$

Note that the maximization of $l_{\boldsymbol{X}}$ still requires to evaluate the expectation of the label function $z$ for given parameters $(\alpha, \boldsymbol{Q})$ by summing over $k^n$ possible labels. To circumvent this problem, one can use the mean-field approximation, which amounts to approximating the posterior distribution $\mathbb{P}(\cdot | \boldsymbol{X} \odot \boldsymbol{A}, \alpha, \boldsymbol{Q})$ by a product distribution. To ensure that this product distribution remains close to the posterior distribution, the objective function is penalized by the Kullback-Leibler divergence of the two distributions. More precisely, the posterior distribution $\mathbb{P}(\cdot | \boldsymbol{X} \odot \boldsymbol{A}, \alpha, \boldsymbol{Q})$ is approximated by a multinomial distribution denoted $\mathbb{P}_\tau$, such that $\mathbb{P}_\tau(z) = \prod_{1 \leq i \leq n} m(z | \tau^i)$, where $m(\cdot | \tau^i)$ is the density of the multinomial distribution with parameter $\tau^i = \left( \tau_1^i, ..., \tau_k^i \right)$, and $\tau = \left( \tau^1, ..., \tau^n \right)$. Then, the variational estimator is defined as

$$\left( \widehat{\alpha}^{VAR}, \widehat{\boldsymbol{Q}}^{VAR}, \widehat{\tau}^{VAR} \right) = \operatorname*{arg\,max}_{\alpha \in \mathcal{A}, \boldsymbol{Q} \in \mathcal{Q}, \tau \in \mathcal{T}} \mathcal{J}_{\boldsymbol{X}}(\boldsymbol{A}; \tau, \alpha, \boldsymbol{Q}) \tag{6}$$

$$\text{for} \quad \mathcal{J}_{\boldsymbol{X}}(\boldsymbol{A}; \tau, \alpha, \boldsymbol{Q}) = \log\left( l_{\boldsymbol{X}}(\boldsymbol{A}; \alpha, \boldsymbol{Q}) \right) - KL\left( \mathbb{P}_\tau(\cdot) || \mathbb{P}(\cdot | \boldsymbol{X} \odot \boldsymbol{A}, \alpha, \boldsymbol{Q}) \right)$$

where $\mathcal{A}$, $\mathcal{Q}$ and $\mathcal{T}$ are the respective parameter spaces for the parameters $\alpha$, $\boldsymbol{Q}$ and $\tau$, $KL$ denotes the Kullback-Leibler divergence between two distributions, and $\boldsymbol{X} \odot \boldsymbol{A}$ denotes the observed entries of $\boldsymbol{A}$. Since for any parameter $(\alpha, \boldsymbol{Q})$, $KL\left( \mathbb{P}_\tau(\cdot) || \mathbb{P}(\cdot | \boldsymbol{X} \odot \boldsymbol{A}, \alpha, \boldsymbol{Q}) \right) \geq 0$, we see that $\exp\left( \mathcal{J}_{\boldsymbol{X}}(\boldsymbol{A}; \tau, \alpha, \boldsymbol{Q}) \right)$ provides a lower bound on $l_{\boldsymbol{X}}(\boldsymbol{A}; \alpha, \boldsymbol{Q})$.

The expectation - maximization (EM) algorithm derived in [48] can be used to iteratively compute the variational estimator. For details, see Appendix C. Since this algorithm is not guaranteed to converge to a global maximum, it should be initialized with care, by using, for example, a first clustering step. This solution is implemented in the package `missSBM`.

Statistical guarantees for the variational estimator obtained in [13, 7, 37] establish that maximizing $\max_{\tau \in \mathcal{T}} \mathcal{J}_{\boldsymbol{X}}(\boldsymbol{A}; \tau, \alpha, \boldsymbol{Q})$ is equivalent to maximizing $l_{\boldsymbol{X}}(\boldsymbol{A}; \alpha, \boldsymbol{Q})$, and that the estimator obtained by maximizing $l_{\boldsymbol{X}}(\boldsymbol{A}; \alpha, \boldsymbol{Q})$ converges to the true parameters $(\alpha^*, \boldsymbol{Q}^*)$. This in turn implies that $(\widehat{\alpha}^{VAR}, \widehat{\boldsymbol{Q}}^{VAR})$ also converges to $(\alpha^*, \boldsymbol{Q}^*)$. Note that these results do not provide guarantees on the recovery of the true labels $z^*$ or on the matrix of connection probabilities $\boldsymbol{\Theta}^*$. In order to estimate $\boldsymbol{\Theta}^*$, we first define the label estimator $\widehat{z}^{VAR}$ using the minimizer of the objective function (6):

$$\forall i \leq n, \ \widehat{z}^{VAR}(i) \triangleq \operatorname*{arg\,max}_{a \leq k} \left( \widehat{\tau}^{VAR} \right)_a^i. \tag{7}$$

Once we have estimated the community labels using (7), we replace the estimator $\widehat{\boldsymbol{Q}}^{VAR}$ of the matrix of connection probabilities by the empirical mean estimator:

$$\forall a < k \text{ and } b < k, \widehat{\boldsymbol{Q}}_{ab}^{ML-VAR} \triangleq \frac{\sum_{(i,j) \in (\widehat{z}^{VAR})^{-1}(a,b)} \boldsymbol{X}_{ij} \boldsymbol{A}_{ij}}{\sum_{(i,j) \in (\widehat{z}^{VAR})^{-1}(a,b)} \boldsymbol{X}_{ij}}$$

and define $\widehat{\boldsymbol{\Theta}}^{VAR}$ as $\quad \widehat{\boldsymbol{\Theta}}_{i\neq j}^{VAR} = \widehat{\boldsymbol{Q}}_{\widehat{z}^{VAR}(i),\widehat{z}^{VAR}(j)}^{ML-VAR}, \quad \widehat{\boldsymbol{\Theta}}_{ii}^{VAR} = 0. \qquad (8)$

We will show respectively in Theorems 2 and 3 that this new estimator $\left(\widehat{z}^{VAR}, \widehat{\boldsymbol{Q}}^{ML-VAR}\right)$ is minimax optimal for dense networks with missing observations as well as for sparse networks. The simulation study provided in Section 4 reveals that this estimator also has good performances in practice.

## 3.2 Convergence rates of variational approximation to the maximum likelihood estimator

In this section, we show the asymptotic equivalence of $\widehat{z}^{VAR}$ and $\widehat{z}$, where

$$(\widehat{\boldsymbol{Q}}, \widehat{z}) \in \underset{\boldsymbol{Q}\in\mathcal{Q}, z\in\mathscr{Z}_{n,k}}{\arg\max} \mathcal{L}_{\boldsymbol{X}}(\boldsymbol{A}; z, \boldsymbol{Q}) \qquad (9)$$

is the maximum likelihood estimator. More precisely, we show that, with large probability, there exists a permutation $\sigma$ of $\{1,...,k\}$ such that $\left(\widehat{z}^{VAR}(\sigma(a))\right)_{a\leq k} = (\widehat{z}(a))_{a\leq k}$ and $\left(\widehat{\boldsymbol{Q}}_{\sigma(a),\sigma(b)}^{ML-VAR}\right)_{a,b\leq k} = \left(\widehat{\boldsymbol{Q}}_{a,b}\right)_{a,b\leq k}$. When this hold, the error of the tractable estimator $\left(\widehat{z}^{VAR}, \widehat{\boldsymbol{Q}}^{ML-VAR}\right)$ matches the minimax optimal error rate. These results are established under the following assumptions:

A.1 There exists $c > 0$ and a compact interval $C_{\boldsymbol{Q}} \subset (0,1)$ such that $\mathcal{A} \subset [c, 1-c]^k$ and $\mathcal{Q} \subset C_{\boldsymbol{Q}}^{k\times k}$;

A.2 The true parameters $\alpha^*$ and $\boldsymbol{Q}^*$ lie respectively in the interior of $\mathcal{A}$ and $\mathcal{Q}$;

A.3 The coordinates of $\alpha^*\boldsymbol{Q}^*$ are pairwise distinct.

Note that Assumption A.2 and A.3 are standard. Assumption A.2 requires that the true parameters lie in the interior of the parameter space, which is classical in parametric estimation. In the most simple case, the parameters $\alpha^*$ and $\boldsymbol{Q}^*$ lie respectively in the interior of sets $\mathcal{A}$ and $\mathcal{Q}$ of the form $\mathcal{A} = [c, 1-c]^k$, $\mathcal{Q} = [c', 1-c']_{sym}^{k\times k}$, for some $c, c' \in (0, 1/2)$. Assumption A.3 ensures the identifiability of stochastic block model parameters. Then, under the assumption that $p = \omega\left(n/\log(n)\right)$, strong recovery of the labels is possible. Assumption A.1 is more restrictive, as it implies that the network is dense. This assumption will be relaxed in Theorem 3, where we consider sparse stochastic block models such that $\boldsymbol{Q}^* = \rho_n\boldsymbol{Q}^0$ for some fixed $\boldsymbol{Q}^0$ and some decreasing, sparsity inducing sequence $\rho_n$.

The following Theorem shows the minimax optimality of the tractable estimator $\widehat{\boldsymbol{\Theta}}^{VAR}$ under assumptions A.1 - A.3.

**Theorem 2.** *Assume that $\boldsymbol{A}$ is generated from a stochastic block model with parameters $(\alpha^*, \boldsymbol{Q}^*)$ satisfying assumptions A.1 - A.3. Then, $\mathbb{P}\left(\widehat{z}^{VAR} \sim \widehat{z}\right) \to 1$ when $n \to \infty$. Moreover, there exists a constant $C_{\boldsymbol{Q}^*} > 0$ depending on $\boldsymbol{Q}^*$ such that*

$$\mathbb{P}\left(\left\|\boldsymbol{\Theta}^* - \widehat{\boldsymbol{\Theta}}^{VAR}\right\|_2^2 \leq \frac{C_{\boldsymbol{Q}^*}\left(k^2 + n\log(k)\right)}{p}\right) \underset{n\to\infty}{\to} 1.$$

Let us now discuss the extension of Theorem 2 to the case of sparse networks. To avoid technicalities, we will consider the case when the network is fully observed. We will also assume that the proportions of different communities are held constant, while the probabilities of connections between communities may decreases at rate $\rho_n$. That is, the parameters $(\alpha^*, \boldsymbol{Q}^*)$ verify the following assumptions:

A.4 $\alpha^* = \alpha^0$ for some fixed $\alpha^0$ such that $\alpha_a^0 > 0$ for any $a \in \{1,...,k\}$

A.5 $\boldsymbol{Q}^* = \rho_n\boldsymbol{Q}^0$ for some fixed $\boldsymbol{Q}^0 \in (0,1)^{k\times k}$ such that $\sum_{a,b=1}^k \alpha_a^0\alpha_b^0\boldsymbol{Q}_{ab}^0 = 1$

Assumption A.5 relaxes Assumption A.1 and allows us consider sparse networks. The normalization constraint $\sum_{1\leq a,b\leq k} \alpha_a^0\alpha_b^0\boldsymbol{Q}_{ab}^0 = 1$ ensure the identifiability of the parameters $(\boldsymbol{Q}^0, \rho_n)$ (see [7]). In the following, we denote by $\mathcal{Q}$ the set of parameters $(\alpha, \boldsymbol{Q})$ verifying Assumptions A.4 and A.5.

The following theorem provides the analogous of Theorem 2 in the case of fully observed sparse networks. It is obtained by combining Propositions 2 and 3 in [19]:

**Theorem 3.** *Assume that $\boldsymbol{A}$ is fully observed, and is generated from a stochastic block model with parameters $(\alpha^*, \boldsymbol{Q}^*)$ satisfying Assumptions A.4 and A.5, such that $\boldsymbol{Q}^0$ has no identical columns and the sparsity inducing sequence $\rho_n$ satisfies $\rho_n \gg \log(n)/n$. Then, $\mathbb{P}\left(\widehat{z}^{VAR} \sim \widehat{z}\right) \to 1$ when $n \to \infty$. Moreover, there exists a constant $C_{\boldsymbol{Q}^0} > 0$ depending on $\boldsymbol{Q}^0$ such that*

$$\mathbb{P}\left(\left\|\boldsymbol{\Theta}^* - \widehat{\boldsymbol{\Theta}}^{VAR}\right\|_2^2 \leq C_{\boldsymbol{Q}^0}\rho_n\left(k^2 + n\log(k)\right)\right) \underset{n\to\infty}{\to} 1. \tag{10}$$

Theorems 2 and 3 establish that the variational estimator $\widehat{\boldsymbol{\Theta}}^{VAR}$ is minimax optimal for both the estimation of dense networks with observations missing uniformly at random, and sparse networks. For proofs and discussion see Appendix A.

## 4 Numerical Results

### 4.1 Synthetic data

In this section we provide a simulation study of the performances of the maximum likelihood estimator defined in (8), and compare it to the variational estimator defined in [48] and implemented in the package `missSBM`, as well as to the Universal Singular Value Thresholding estimator introduced in [25] and implemented in the package `softImpute`. The results are reported in Figure 1. Thorough descriptions of the simulation protocols are provided in the Appendix.

**Dense stochastic block model**    First, we evaluate the empirical performances of the variational approximation of the maximum likelihood estimator defined in (8) on dense stochastic block models. We estimate the matrix of probabilities of connections, and we compare our estimator with the estimator given by the methods missSBM and softImpute. The quality of the inference is assessed by computing the squared Frobenius distance between the estimators and the true matrix of connection probabilities $\boldsymbol{\Theta}^*$.

We consider three types of three-communities stochastic block model. The first model, given by $(\alpha^{assort.}, \boldsymbol{Q}^{assort.})$, provides a simple assortative network, where individuals are more connected with people from their communities than with other individuals. On the contrary, the second model, given by $(\alpha^{disassort.}, \boldsymbol{Q}^{disassort.})$, is disassortative: individuals are more connected with individuals from outside of their communities. Both the assortative and disassortative models have balanced communities. The third model considered, given by $(\alpha^{mix.}, \boldsymbol{Q}^{mix.})$, exhibits neither assortativity nor disassortativity, and the communities are unbalanced. We introduce missing data by observing each entry of the adjacency matrix independently with probability 0.5.

The variational approximation to the maximum likelihood estimator defined in (8) outperforms the softImpute method across all models and all number of nodes. Its error is equivalent to that of the oracle estimator with hindsight knowledge of the true label function $z^*$ when the network is a few hundred nodes large. Interestingly, our estimator also outperforms the variational estimator implement in the package `missSBM`. We underline however that the primary focus of the missSBM method is to infer the parameters $(\alpha^*, \boldsymbol{Q}^*)$.

Additional experiments illustrating the strong consistency of the variational estimator canbe found in Appendix B.2.

**Sparse stochastic block model**    Next, we investigate the behaviour of our estimator on increasingly sparse networks. We consider a three-communities assortative stochastic block model of 500 nodes with balanced communities, and 50% missing values. The probabilities of connections are given by $\boldsymbol{Q}^* = \rho\boldsymbol{Q}^0$, where $\rho$ is a parameter controlling the sparsity, which ranges from 0.05 to 1. We compare the performance of the variational approximation to the maximum likelihood estimator to that of the methods softImpute and missSBM. We also compare these estimators to the trivial estimator with all entries equal to the average degree divided by the number of nodes. The error is measured as the squared Frobenius distance between the estimator and the matrix $\boldsymbol{\Theta}^*$ divided by $\rho^2$.

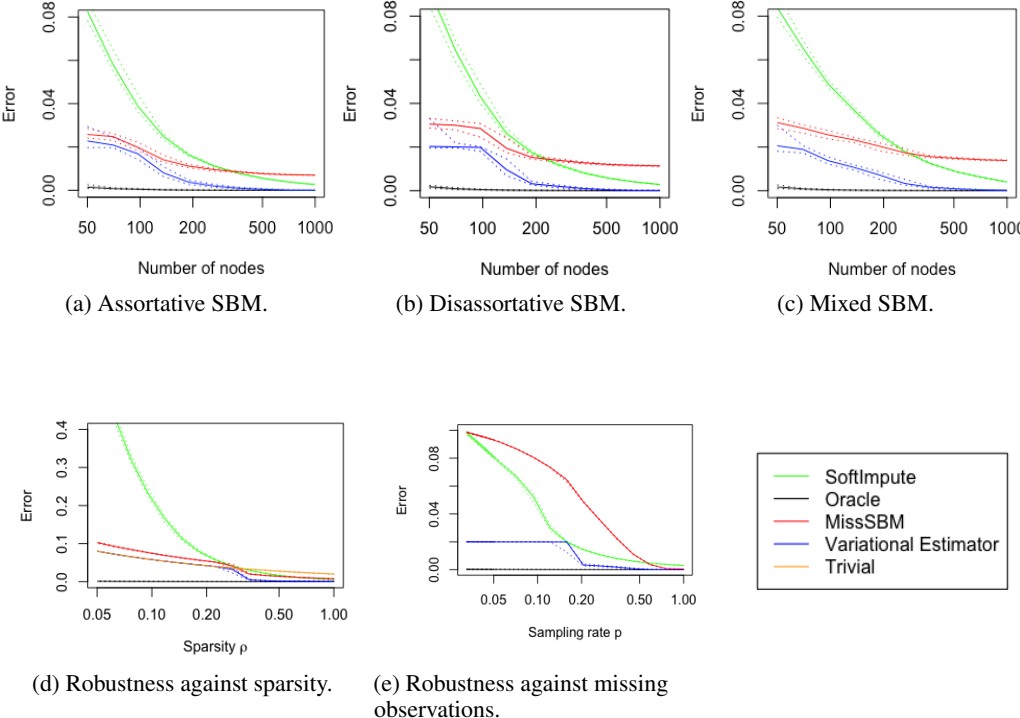

Figure 1: Top : Error of connection probabilities estimation as a function of the number of nodes (top left : assortative SBM with balanced communities; top middle : disassortative SBM with balanced communities; top right : mixed SBM with unbalanced communities) or of the sparsity parameter $\rho$ (bottom left) and of the sampling rate $p$ (bottom right). We compare the variational approximation to the maximum likelihood estimator (in blue) to that of `missSBM` (in red), that of `softImpute` (in green), that of the oracle estimator with knowledge of the label $z^*$ (in black), and that of the trivial estimator with entries equal to the empirical average degree divided by the number of nodes (orange, bottom only). The full lines indicate the median respectively of the mean squared error (top and bottom right) and of the mean squared error divided by the sparsity parameter $\rho$ (bottom left) of the estimators over 100 repetitions, while the dashed lines indicate its 25% and 75% quantiles.

As the network sparsity increases, the clustering of the nodes becomes more difficult. The normalized error of the estimator $\widehat{\Theta}^{VAR}$ increases up to a threshold corresponding to the normalized error of the trivial estimator with all entries equal to the empirical degree, divided by the number of nodes. Note that when considering very sparse networks, with $\rho \ll \log(n)/n$, it is known that the trivial estimator with entries equal to the empirical mean degree is minimax optimal (see, eg, [29])). Thus, the estimator enjoys relatively low error rates in both high and low signal regime. By contrast, the normalized error of the softImpute method diverges as the network becomes increasingly sparse.

**Stochastic block model with missing observations**    To conclude our simulation study, we evaluate the robustness of the methods against missing observations. We consider a three-communities assortative stochastic block model with balanced communities and $500$ nodes. We increase the proportion of missing observations, and we compare the performance of the variational approximation to the maximum likelihood estimator to that of the methods softImpute and missSBM. The error is measured as the squared Frobenius distance between the estimator and the matrix $\Theta^*$.

As the sampling rate $p$ decreases, the clustering becomes impossible and the error rate of the estimator $\widehat{\Theta}^{VAR}$ increases up to that of the trivial estimator obtained by averaging the observed entries of the

adjacency matrix. By contrast, the methods softImpute and missSBM lack robustness against missing observations, and their error diverges as the number of missing observations increases.

## 4.2 Analysis of real networks

### 4.2.1 Prediction of interactions within a elementary school

We apply our algorithm to analyze a network of interactions within a French elementary school collected by the authors of [47]. The network records durations of physical interactions occurring within a primary school between 222 children divided into 10 classes and their 10 teachers over the course of two consecutive days; this dataset was collected using a system of sensors worn by the participants. We consider that an interaction has occurred if the corresponding duration is greater than one minute. If an interaction of less than one minute is observed, we consider that this observation may be erroneous, and treat the corresponding data as missing. By doing so, we remove respectively 11 and 13% of the observations on Day 1 and Day 2.

The graphs of interactions recorded during Day 1 and Day 2 can be considered as two outcomes of the same random network model characterized by the matrix of connection probabilities $\Theta^*$. In this spirit, we use the observations collected on Day 1 estimate the matrix $\Theta^*$, and evaluate those estimators on the network of interactions corresponding to Day 2. We note that the network of interactions for Day 1 has rather homogeneous degrees (the maximum degree is 41 and the minimum degree is 5, while the mean degree is 20). Moreover, it exhibits a strong community structure. Therefore, we expect the networks of interactions to be well approximated by a stochastic block model.

We compare the performance in terms of link prediction of the estimator $\widehat{\Theta}^{VAR}$ defined in (8) to that of the method missSBM, and that of the method softImpute. In this last method, we set the penalty to 0, and we choose the rank of the estimator to be equal to the number of communities, which is estimated according to the Integrated Likelihood Criterion. We also compare these methods to the naive persistent estimator $\widehat{\Theta}^{naive}$ given by $\widehat{\Theta}^{naive}_{ij} = 1$ if an interaction between $i$ and $j$ has been recorded on Day 1, $\widehat{\Theta}^{naive} = 0$ if no such interaction has been recorded, and $\widehat{\Theta}^{naive}_{ij} = d/n$ if the information is missing, where $d$ is the average degree of the graph for Day 1. Table 1 present the error of the different estimators, measured as the squared Frobenius distance between the adjacency matrix of Day 2 and its predicted value, divided by the squared Frobenius norm of the adjacency matrix of Day 2 (i.e, the error of the trivial null estimator).

| Estimator | $\widehat{\Theta}^{VAR}$ | $\widehat{\Theta}^{missSBM}$ | $\widehat{\Theta}^{SVT}$ | $\widehat{\Theta}^{naive}$ |
|---|---|---|---|---|
| $\|\boldsymbol{X} \odot (\boldsymbol{A} - \widehat{\Theta})\|_2^2 / \|\boldsymbol{X} \odot \boldsymbol{A}\|_2^2$ | 0.312 | 0.317 | 0.357 | 0.541 |

Table 1: Link prediction error on the network of interactions within a primary school.

The variational method predicts most accurately the interactions on Day 2. It is closely followed by the estimator provided by the package missSBM. By contrast to the simulation study, the reduction in error when using the new estimator is moderate : the error of $\widehat{\Theta}^{VAR}$ is respectively 1.4% and 12.4% smaller than that of $\widehat{\Theta}^{missSBM}$ and $\widehat{\Theta}^{softImpute}$. In addition, the precision-recall curve presented in the Appendix indicates that no estimator is better across all sensitivity levels. Interestingly, the naive estimator obtains a high error, which suggests a certain versatility in the children's behaviour.

### 4.2.2 Network of co-authorship

Finally, we use variational approximation to predict unobserved links in a network of co-authorship between scientists working on network analysis, first analysed in [43]. We discard the smallest connected components (with less than 5 nodes), and we obtain a network of 892 nodes. By contrast to the network of interaction in an elementary school, the network of co-authorship is quite sparse, and presents heterogeneous degrees: the average number of collaborators is 5, while the maximum and minimum number of collaborators are respectively 37 and 1.

In order to obtain unbiased estimates of the error of the estimators $\widehat{\Theta}^{VAR}$, softImpute, and missSBM, we introduce 50% of missing values in the dataset. We train the three estimators on the observed

entries of the adjacency matrix, and we use the unobserved entries to evaluate their imputation error. Table 2 present the mean imputation error of the different estimators over 100 random samplings, measured in term of squared Frobenius error and normalized by the squared Frobenius norm of the adjacency matrix of the remaining entries (i.e, the error of the null estimator). Here again, the

| Estimator | $\widehat{\Theta}^{VAR}$ | $\widehat{\Theta}^{missSBM}$ | $\widehat{\Theta}^{SVT}$ |
|---|---|---|---|
| $\|(1-X)\odot(A-\widehat{\Theta})\|_2^2/\|(1-X)\odot A\|_2^2$ | 0.857 | 0.869 | 0.894 |

Table 2: Imputation error of the estimators on the network of co-authorship.

variational approximation to the maximum likelihood estimator obtains the best performance. The precision-recall curves of these methods, included in the Appendix, indicates that this new estimator is preferable across almost all sensitivity levels. We underline however that the errors in term of Frobenius norm of the three estimators are close, and relatively high. This comes as no surprise, as the high sparsity of the network causes the link prediction problem to be difficult.

## 5 Conclusion

In this work, we have introduced a new tractable estimator based on variational approximation of the maximum likelihood estimator. We show that it enjoys the same convergence rates as the maximum likelihood estimator, and that it is therefore minimax optimal. Our simulation studies reveal the advantages of our estimator over current methods. In particular, they highlight its robustness against network sparsity and missing observations. Our results pave the way for analysing variational approximations of more general structured network models such as the latent block model.

## Broader Impact

This paper proposes a new method for network analysis which is theoretically optimal, and has empirical advantages compared to current algorithms. It can be useful to researchers and practitioners with applications, e.g., in biology [40], ethnology [48] and ecology [5]. We acknowledge that improvements in network analysis and link prediction can have negative downstream societal impacts such as contributing to community-based polarization of opinion, or improving online surveillance.

## Acknowledgments

We thank the anonymous referees for their helpfull comments.

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
