# OpenReview forum: "Optimality of variational inference for stochasticblock model with missing links"
_NeurIPS.cc/2021/Conference — NeurIPS 2021 Poster_

### Official Review · Reviewer_1S7t · 2021-07-14

**Rating:** 6
**Confidence:** 3

**Summary:**

The main contribution of this paper is to propose and analyze a variational approximation (VA) method for the inference of the SBM. The main result is to establish that VA leads to a tractable and minimax optimal algorithm, specifically the latter is based on the EM principle with a variational approximation scheme. Numerical experiments are also performed to verify the theory.

**Limitations And Societal Impact:**

There isn't any potential negative societal impact. The limitation/suggestions for improvement have been discussed in the main review above.

**Main Review:**

Overall, the paper is well organized and the results are useful to this community. However, the reviewer has several concerns related to the technical novelty, algorithm implementation, as follows:

- The study motivated the variational approximation approach by the need for complexity reduction and the original MLE problem is replaced by the VA problem in line 170 of Sec. 3.1, which is later on tackled by the EM algorithm. Several issues arise here:

-- The VA problem has taken the model distribution $P_\tau$ as the multinomial distribution to approximate $P(.|X.A, \alpha, Q)$. The reviewer wonders if this is the only model distribution to use for the algorithm? Does such choice of the model distribution affect the minimax optimal performance of the proposed estimator? Currently, the latter choice is not discussed in the paper.

-- It is mentioned in the study that the EM algorithm is initialized using a clustering step such as "missSBM". Furthermore, the analysis in this paper (in the appendix) seems to have assumed that a *global optimizer* is obtained by the EM. The reviewer is concerned about how such global optimality condition can be enforced using the existing algorithm, e.g., is there any guarantee on global optimality when using initialization step such as "missSBM"?

-- In the numerical simulation, the study has only compared the EM algorithm (which is a VA of the MLE) to missSBM and softImpute. The reviewer wonders how is the performance of the true MLE?

- The reviewer notices that a number of the intermediate steps in the proof followed from [34]. Upon checking [34], it is also found that the latter paper has proposed a VA method for the inference of the SBM similar this paper. May I know what is the main technical novelty of the current paper with respect to [34]?

Moreoever, in Sec 3.2 on the the convergence rates results, the compact sets ${\cal A}, {\cal Q}$ in A1, A2 have not been clearly defined. It may be helpful to give examples for these sets.

- As a minor point, it should be noted that the paper has not followed strictly the official format of an NeurIPS submission (e.g., with a smaller font size).

**--- Post Rebuttal ---**

I have read the authors response, which has sufficiently addressed my previous concern. I have updated my score accordingly.

**Time Spent Reviewing:**

5

---

> ### Author Response · Authors · 2021-08-10
> **Response to Reviewer 1S7t**
>
> We thank the reviewer for raising interesting questions and pointing out aspects of our manuscript which we will clarify in a revised version.
>
>  Variational methods are widely used because the variational approximation of the posterior distribution allows to design computationally efficient algorithms for problems that would otherwise not be tractable. While a line of works focuses on studying variational methods in a unified framework, and considers in this spirit a large class of distributions to be used as variational approximations to the posterior distribution, other works, including ours, focus on specific problems and aim at characterizing the statistical properties of the variational approximation given by a distribution adapted to the problem at hand. For example, in the Stochastic Block Model, using a multinomial distribution to approximate the posterior distribution of the label is particularly convenient, as it leads to a tractable implementation of the Expectation-Maximization algorithm. For this reason, mean field approximation is overwhelming in the literature studying variational inference for the SBM (in fact, we are not aware of the existence of a variational algorithm for estimating parameters in the stochastic block model that would rely on an other distribution). Our proofs rely on the use of the multinomial distribution and cannot be currently used to conclude on the optimality of other variational approximations.
>
> Our theoretical results are obtained for an estimator based on the true minimizer of the variational objective function rather than on the output of the Expectation-Maximization algorithm. This algorithm, first introduced by Daudin et al. (2008) [1] for fully observed networks and then extended by Tabouy et al. (2020) [2] to the case of missing observations, is only included in our article for the sake of completeness. We apologize if this point was not clear enough, and we will clarify it in the revised version of the manuscript.
> Most works on variational inference, including ours, aim at establishing guarantees for the true minimizer of the variational objective function, and on comparing it to the true maximum likelihood estimator. In those settings obtaining guarantees on the convergence of the algorithm itself remains in the vast majority of cases an open question. To the best of our knowledge, the only endeavors in this direction have focused on the convergence of algorithms designed for a restrictive class of fully observed, two-parameters, assortative SBM : these algorithms assume that the probability of connections within a community, $p$ and the probability of connections between communities, $q$, are the same across all communities, and that $p>q$ (see, e.g.,  Zhang et al. (2020) [3] and Sarkar et al (2021) [4]).
>
>
> It would indeed be interesting to compare the empirical performances of the EM algorithm used in our study to that of the true maximum likelihood estimator. Unfortunately, to compute the maximum likelihood estimator, one must maximize the objective function (3) over a number of labels growing as $3^n$, where $n$ is the number of nodes. It can therefore not be computed, even for a moderate number of nodes.
>
>
> The current paper establishes the minimax optimality of variational estimation in the SBM with missing observations. This problem is strongly related to that of link prediction, which is very important in applications. Because of the missing observations, estimating the parameters requires   new techniques of proofs. For example, we must use new techniques to compare the restricted and unrestricted maximum likelihood estimators. We also establish as an intermediate result the strong consistency of the maximum likelihood estimator for the conditional SBM (in the full observation setting, this result is a direct consequence of Bickel et al. (2009) [5]). Similarly, the proof of Theorem 3 relies heavily on the fact that the likelihood function at the parameters $\alpha^*, \mathbf{Q}^*$ and the profile likelihood function at the parameters  $z^*, \alpha^*, \mathbf{Q}^*$ are asymptotically equivalent, which is a direct consequence of Lemma 3 in Bickel et al (2013) [6]. This result does not hold under missing observations, and we develop new arguments to prove the strong consistency of the variational estimate of the labels.
>
>
>   An other important contribution of this paper is the illustration of the behavior of our VA estimator by simulation studies. They show that our estimator has good empirical performances, even in difficult settings (where the networks becomes very sparse, or where the number of missing observations is important).
>
>
> We thank the reviewers for their suggestions on how to improve the clarity of our papers. In the revised version of the present paper, we will add to the discussion following the assumptions (A.1) - (A.3) the following sentence : “In the most simple case, the parameters $\alpha^*$ and $\mathbf{Q}^*$ lie respectively in the interior of sets $\mathcal{A}$ and $\mathcal{Q}$ of the form $\mathcal{A} = [c, 1-c]$, $\mathcal{Q} = [c’, 1-c’]^{k \times k}_{sym}$ for some $c, c’$ in $(0, 1/2)$.”
>
>
> We most sincerely apologize for unintentionally setting the font small, which we had not noticed until now. Upon inspection, it appears to have been caused by the command used to create keywords.
>
> [1] Jean-Jacques Daudin, Franck Picard, and Stephane Robin. "A mixture model for random graph". Statistics and Computing, 18:173–183, 2008.
> [2] Timothee Tabouy, Pierre Barbillon, and Julien Chiquet. "Variational inference for stochastic block models from sampled data". Journal of the American Statistical Association, 115(529):455–466, 2020.
> [3] Anderson Y. Zhang, Harrison H. Zhou. "Theoretical and computational guarantees of mean field variational inference for community detection." The Annals of Statistics, 48(5) 2575-2598, 2020
> [4]Sarkar, P., Wang, Y., & Mukherjee, S.S. (2021). When random initializations help: a study of variational inference for community detection. J. Mach. Learn. Res., 22, 22:1-22:46.
> [5] Peter J. Bickel and Aiyou Chen. A nonparametric view of network models and newman–girvan and other modularities. Proceedings of the National Academy of Sciences, 106(50):21068–21073, 2009.
> [6] Peter Bickel, David Choi, Xiangyu Chang, and Hai Zhang. Asymptotic normality of maximum likelihood and its variational approximation for stochastic blockmodels. The Annals of Statistics, 41(4):1922 – 1943, 2013.

---

> > ### Public Comment · ~Jiajun_Liang2 · 2022-01-06
> > **Question on the dimension of $\alpha^*$**
> >
> > Shouldn't $\alpha^*$ be a $k$ by $1$ vector?

---

> > > ### Public Comment · ~Solenne_Gaucher1 · 2022-01-06
> > > **Yes, sorry for the typo**
> > >
> > > Indeed, there is a typo here. It should say "In the most simple cases [...] $\mathcal{A} = [c, 1-c]^k$ ". It has been corrected in the manuscript. Thank you for pointing it out!

---

### Official Review · Reviewer_FvDk · 2021-07-14

**Rating:** 7
**Confidence:** 4

**Summary:**

This paper considers estimation of the parameters of a stochastic block model (community membership and probability parameters) using a variational approach. Both the censored, dense model and the sparse model are considered. Experiments on real and synthetic data compare the estimator to another variational estimator, as well as another estimator based on Universal Singular Value Thresholding.

**Main Review:**

I don’t see much of a novel contribution in this work. The variational estimator is in large part already present in Reference 17. The only change is accounting for the missing observations by including the X_{ij} variables. Theorems 1 and 3 are cited from prior work, while Theorem 2 seems to follow from Proposition 2 in Reference 17 (see discussion below Proposition 2 in Reference 17). Given that the estimate of the communities is accurate, then Theorem 2 follows from Theorem 1.

Moreover, it seems that the main effort is estimation of the community assignments. From this estimate, the estimator of the matrix of edge probabilities takes on a simple form. There will be some regimes where finding the community assignments exactly is possible via existing methods simpler than the variational method proposed, and this should be acknowledged. This applies to both the standard and censored models.

The paper claims optimality, so there should be justification of the optimal rates of the estimators.

Finally, the font size appears to have been set small.

Edit: Score updated in light of discussion.


**Time Spent Reviewing:**

4

---

> ### Author Response · Authors · 2021-08-10
> **Response to Reviewer FvDk**
>
> The reviewer doesn't see "much of a novel contribution in this work": we strongly disagree with this statement. Our main contributions are:
> 1. We prove that there is no computational gap in problem of link prediction for SBM providing an estimator that is minimax optimal and computationally feasible (this question had been open for a while)
> 2. We provide numerical experiments for this new algorithm to compare it to those previously proposed in the literature.
>
> The reviewer mentions that "The only change is accounting for the missing observations". It is well known that this "only change" raises an important and challenging problem of link prediction exactly as adding missing observations to a low rank matrix raised the problem of matrix completion (that is much harder than the problem of low rank matrix estimation).
>
> Theorem 2 doesn't follow from Proposition 2 in Reference 17 (as a look at its proof in the appendix will show). The discussion below Proposition 2 in Reference 17 doesn't give a clue on the actual proof of Theorem 2 which, as our paper shows, is quite technical and involved. Contrary to what is stated in the report, Theorem 2 does not "just follow" from Theorem 1 either.
>
> As noted by the reviewer, our results hold in a regime where exact recovery of the labels is possible, as indicated by our proofs, however we don't know any « simpler » algorithm that would allow to exactly recovery of the labels in our setting. We underline that our algorithm is studied assuming a class of models more general than the  fully observed, balanced, assortative, two-parameters SBM predominant in articles which focus on sharp recovery of the labels (where the matrix $\mathbf{Q}$ has diagonal entries all equal to a parameter $p$ and of-diagonal entries all equal to $p$, and $p>q$). For the more realistic case where $\mathbf{Q}$ is only constrained to be symmetric and to have positive entries, and some observations on the network are missing, we are not aware of simpler algorithms that would allow for perfect recovery of the labels. We would be gratefull if the reviewer could provide us with references on how to exactly recover the labels in this setting.
>
> We emphasize once again that the main point of the paper is to show that there is no computational gap in the problem of link prediction for SBM. We not only claim optimality but we prove it. The upper bound is given by Theorems 2. This upper bound matches the lower bounds established in Klopp et al. (2017) [1] and Gao et al. (2016) [2] (see the discussion after Theorem 1). Moreover our simulation studies show that our estimator has good empirical performances, even in difficult settings (where the networks becomes very sparse, or where the number of missing observations is important).
>
> We most sincerely apologize for unintentionally setting the font small. Upon inspection, it appears to have been caused by the command used to create keywords.
>
> [1] Olga Klopp, Alexandre B. Tsybakov, and Nicolas Verzelen. "Oracle inequalities for network models and sparse graphon estimation." The Annals of Statistics, 45(1):316 – 354, 2017.
>
> [2] Chao Gao, Yu Lu, Zongming Ma, and Harrison H. Zhou. 2016. "Optimal estimation and completion of matrices with biclustering structures". J. Mach. Learn. Res. 17, 1 (January 2016), 5602–5630.

---

> > ### Comment · Reviewer_FvDk · 2021-08-10
> > **Discussion**
> >
> > Thank you for your comments. Re-examining your paper raised a few points:
> > - Thank you for clarifying the novelty and significance. It would be helpful if you could indicate which new ideas are used to prove Theorem 2. If you don't have space for this, please include a discussion of the technical novelty in the appendix, and refer to it in the main paper. My misunderstanding about novelty came from taking only a cursory look at Reference 17.
> > - The proof of Theorem 2 states that \hat{z} \tilde z^{\star} with high probability, which makes it seem that the assumptions of Theorem 2 place the model in the exact recovery regime. Please make this explicit.
> > - The scaling of p is a bit ambiguous. You state that the reveal probability is \gg \log(n)/n. Does this mean \omega(\log(n)/n)? This is higher than the typical scaling of \Theta(\log(n)/n), and should be acknowledged. Do you think your results would hold under the typical scaling? Some discussion would be valuable. Of course, when p = \Theta(\log(n)/n), there are parameter settings under which exact recovery in the censored SBM is information-theoretically impossible, so we cannot have \hat{z}^{VAR} \tilde z^{\star} whp.
> > - You are right that the setting you consider is quite a bit more general than other works on exact recovery in the censored block model. Still, it would be nice to connect to the literature on censored SBMs. For example, you can see the following recent paper and references therein:
> > Dhara, S., Gaudio, J., Mossel, E., & Sandon, C. (2021). Spectral Recovery of Binary Censored Block Models. arXiv preprint arXiv:2107.06338
> > - Small comment: please define \tilde, as in \hat{z}^{VAR} \tilde \hat{z}. I assumed x \tilde y means that x and y define the same community partition, up to relabelling of the communities. Is that right?
> > -Please indicate how you will fit in 9 pages after increasing the font size.

---

> > > ### Author Response · Authors · 2021-08-13
> > > **Response to further questions of Reviewer FvDk**
> > >
> > > We thank the reviewer for his/her suggestions.
> > >
> > >   - We will indicate the new ideas used to prove Theorem 2 in the Appendix and will refer to it in the main paper. In this part, we will clarify that because of missing links we must use new techniques to compare the restricted and unrestricted maximum likelihood estimators. We also need to establish as an intermediate result the strong consistency of the maximum likelihood estimator for the conditional SBM (in the full observation setting, this result is a direct consequence of Bickel et al. (2009) [1]). Similarly, the proof of Theorem 3 relies heavily on the fact that the likelihood function at the parameters $\alpha^*, \mathbf{Q}^*$ and the profile likelihood function at the parameters  $z^*, \alpha^*, \mathbf{Q}^*$ are asymptotically equivalent, which is a direct consequence of Lemma 3 in Bickel et al (2013) [2]. This result does not hold under missing observations, and we develop new arguments to prove the strong consistency of the variational estimate of the labels.
> > >
> > > -  We will add to the discussion on the assumptions of Theorem 2 that they imply that exact recovery of the label function is possible;
> > >
> > > - As noted by the reviewer, the probability $p$ of observing an entry of the adjacency matrix scales as $\omega(\log(n)/n)$ (this point will be clarified in the revised version of the manuscript). This ensures that recovery of the labels is possible as long as the model is identifiable (i.e., either under assumption A.3 or under the assumption that $Q$ has no two identical columns), without imposing further assumptions on the distance between the connection probabilities of the different communities. On the other hand, we are not so sure that the scaling \Theta(log(n)/n) may be called  "typical" when dealing with the probability of observation of an edge in the network with missing links. Indeed, this scaling is "typical" for the sparsity of the network and in the recent pre-print suggested by the referee [7]  it is shown that this scaling provides the threshold for the exact recovery of the communities in the dense case with two balanced communities. As in the dense case this problem is strongly related to one-bit matrix completion, based on the results from the literature on one bit matrix completion we may conjecture that to go beyond the case considered in [7] we will still need strong additional conditions on the matrix (such as incoherence condition) to be able exactly reconstruct the communities with $p$ which scales as $ \Theta(log(n)/n)$.
> > >
> > > - We thank the reviewer for pointing out  the  very  interesting reference [7]. As it is posterior to our contribution we obviously couldn't cite it but we gladly will add it to the revised version of our paper, as well as references [4]  and [8] which adress the exact recovery problem in the Censored Stochastic Block Model with two balanced communities.
> > >
> > > -   Equivalence of label functions up to a permutation of the communities, denoted by $\sim$, has been defined after the introduction of the stochastic block model; however it would indeed be more convenient to define it in Section 1.2 with the rest  of the notations.
> > >
> > > - Increasing the font size will imply the following minor changes :
> > >
> > > - we have gathered Figures 1 and 2 so as to avoid redundancy in the captions;
> > >
> > > - we have moved the description of the E-M algorithm to the Appendix. This algorithm, which was designed by Daudin et al. (2008) [5] and adapted by Tabouy et al. (2020) [6] to the case of missing observations, was included in our paper for the sake of completeness;
> > >
> > > - we have omitted the definition of our estimator under complete observation of the network, as it can be deduced directly from its definition in the missing observation setting.
> > >
> > > Minor rephrasing of several sentences allow us to comply with the 9-pages limit.
> > >
> > > [1] Peter J. Bickel and Aiyou Chen. A nonparametric view of network models and newman–girvan and other modularities. Proceedings of the National Academy of Sciences, 106(50):21068–21073, 2009.
> > >
> > > [2] Peter Bickel, David Choi, Xiangyu Chang, and Hai Zhang. Asymptotic normality of maximum likelihood and its variational approximation for stochastic blockmodels. The Annals of Statistics, 41(4):1922 – 1943, 2013.
> > >
> > > [3] Mahendra Mariadassou and Timothée Tabouy. Consistency and asymptotic normality of stochastic block models estimators from sampled data. Electronic Journal of Statistics, 14(2):3672 – 3704, 2020.
> > >
> > > [4] E. Abbe, A. S. Bandeira, A. Bracher, and A. Singer. Decoding binary node labels from censored edge measurements: Phase transition and efficient recovery. IEEE Transactions on Network Science and Engineering, 1(1):10–22, 2014.
> > >
> > > [5] Jean-Jacques Daudin, Franck Picard, and Stéphane Robin. A mixture model for random graph. Statistics and Computing, 18:173–183, 06 2008.
> > >
> > > [6] Timothée Tabouy, Pierre Barbillon, and Julien Chiquet. Variational inference for stochastic block models from sampled data. Journal of the American Statistical Association, 115(529):455–466, 2020.
> > >
> > > [7] Dhara et al, Spectral Recovery of Binary Censored Block Models, July 2021
> > >
> > > [8] B. Hajek, Y. Wu, and J. Xu. Exact recovery threshold in the binary censored block model. In 2015 IEEE Information Theory Workshop-Fall (ITW), pages 99–103. IEEE, 2015.

---

> > > > ### Comment · Reviewer_FvDk · 2021-08-13
> > > > **Response to reviewer**
> > > >
> > > > Thank you for the clarifications! In light of my misunderstanding, I have updated my score.

---

### Official Review · Reviewer_UYYv · 2021-07-15

**Rating:** 7
**Confidence:** 3

**Summary:**

Paper provides an approximation to MLE of SBM with missing observations variational via variational inference. Paper provided proofs for statistical optimality and associated simulations showing advantages to existing methods.

**Limitations And Societal Impact:**

Paper mentioned potential negative societal impact. Good for a theoretical work.

**Main Review:**

This is a strong work on the variational inference of SBM. The paper achieved the following
- Connected the variational inference estimator to the MLE, which is believed to be the optimal estimator in the minimax sense. Paper provided the consistency between the two and gave careful mathematical analysis.
- Simulation shows good performance of the proposed method
- An EM algorithm is constructed for approximating the VI estimator. Although the convergence guarantee is not built, paper mentioned techniques on avoiding falling into local maximas

In general, a good contribution to the SBM literature and I'd recommend the conference to accept.

Some points of improvement
- Consistency result between VI and MLE is very interesting. Theorem suggests that both estimators are the same up to a permutation, which is very surprising since when node number goes up, the signal level requirement is high to ensure such consistency. If you could explain a bit more on this that would be helpful
- It would be interesting to see community localization errors (say, ratio of nodes with community labels wrong). In this sense you don't need an oracle (which always gives 0 errors), and the ranges are between 0-1 so it gives sense to cross comparison.

**Time Spent Reviewing:**

2

---

> ### Author Response · Authors · 2021-08-10
> **Response to the review by Reviewer UYYv**
>
> We thank the reviewer for raising very interesting questions.
>
> The ability of both the maximum likelihood estimator and its variational counterpart to perfectly recover all labels is indeed a very interesting fact. Note that while the number of nodes that need to be correctly classified increases, the number of observations per node also increases dramatically (namely, it increases as $p\rho_n n$). The situation where the sparsity of the network is not too important and the average number of observed connections is greater than log(n) corresponds to a strong SNR setting: the signal per node is sufficiently important to recover exactly the labels. Results implying exact recovery of the labels have already been established in this regime for different estimators under more restricted assumptions (e.g., under the assumption that the SBM is symmetric, assortative and has balanced communities, and that the network is fully observed). See, e.g., Abbe et al. (2017) [1] for a review of these results.
>
> While recovery of the labels is not the primary purpose of our estimator, it is indeed interesting to monitor the proportion of misclassified nodes. Our simulations indicate that the average proportion of misclassified nodes indeed decreases to 0 as the number of nodes increases in the models considered. As expected, the classification problem becomes more difficult and the proportion of misclassified nodes increases when the sparsity of the network increases, or when the number of observations decreases. We will include the results of these simulations in the revised version of the manuscript’s appendix.
>
> [1] E. Abbe and C. Sandon. Community detection in general stochastic block models: Fundamental limits and efficient algorithms for recovery. In 2015 IEEE 56th Annual Symposium on Foundations of Computer Science, pages 670–688, 2015.

---

### Official Review · Reviewer_Pcfk · 2021-07-16

**Rating:** 7
**Confidence:** 2

**Summary:**

This paper proves that mean-field variational inference is minimax optimal for parameter estimation of the stochastic block model (SBM) with missing data. The paper establishes this result for both dense networks as well as for (more realistic) sparse networks. The paper further reports the results of experiments on both real and synthetic data that clearly corroborate the paper's main theoretical results.

**Ethical Concerns:**

None.

**Limitations And Societal Impact:**

Yes.

**Main Review:**

This paper contributes a significant morsel of knowledge to what we know about the statistical properties of variational inference. The result is powerfully simple: VB is minimax optimal for the SBM with missing links for both dense and sparse networks. This result should impact directly the statistical networks community but also provide a stepping stone to broader/more general theoretical analysis of variational inference.

I say all of that with the caveat that I am not up-to-date with all of the most recent literature on theoretical analysis of VB and have questions for the authors about how their results relate to a pair of related papers that are not cited in the current version. Can the authors describe the relation of their paper to the papers of Debdeep et al. (2018) and Yang et al. (2020) [2]? The first paper provides "general conditions for obtaining optimal risk bounds for point estimates acquired from mean-field variational Bayesian inference" and gives examples related to the SBM (i.e., latent Dirichlet allocation). The second paper also provides similar kinds of results that establish minimax optimality of variational Bayes under different conditions.

In terms of clarity, the paper is well-written but suffers from some notational headaches. The authors should clean up their notation and employ more aliasing so that, for example, the bounds on sums do not take up the entire equation (see line 128).

Minor comments:
* Add legends to figures 1 and 2.
* Line 60: "Constribution" -> "Contributions"
* Line 95: "citations" -> "citation"

[1] Pati, Debdeep, Anirban Bhattacharya, and Yun Yang. "On statistical optimality of variational Bayes." International Conference on Artificial Intelligence and Statistics. PMLR, 2018.

[2] Yang, Yun, Debdeep Pati, and Anirban Bhattacharya. "$\alpha$-variational inference with statistical guarantees." The Annals of Statistics 48.2 (2020): 886-905.

**Time Spent Reviewing:**

3

---

> ### Author Response · Authors · 2021-08-10
> **Response to the review by Reviewer Pcfk**
>
> We thank the reviewer for pointing out these interesting references, which will be included in the revised version of our manuscript.
>
> Depdeep et al. (2018)[1] study the convergence rates of point estimates obtained using (mean field) variational approximation. Their results apply to rather general models, however their setting is Bayesian in essence. By comparison to our model, they introduce priors over the parameters characterizing the distributions of both the latent variables, and the observations conditionally on these latent variables. In the Stochastic Block Model, this amounts to introducing priors over the parameters $\alpha$, $\mathbf{Q}$. In this setting, the authors show it is asymptotically equivalent to minimize the variational objective function, or the Bayes risk obtained using the variational estimate of the posterior distribution of the parameters. They also obtain bounds on this variational approximation to the Bayes risk under additional assumptions that do not hold in the SBM : more precisely, they assume that there are as many latent variables $z_i$ as observations $Y_i$, and that the pairs $(z_i, Y_i)$ are i.i.d. conditionally on the parameters of the model. Generalizing their results to the SBM is not straightforward, partly because the problem is more complex due to the dependence of the edges on the labels of both nodes.
>
> Moreover, the authors of Depdeep et al. (2018 )[1] focus on the estimation of the parameters $\alpha$, $\mathbf{Q}$. They do not provide any guarantee on the estimation of the latent variables. In this sense, their work is close to that of Bickel. et al. (2013) [2] and Mariadassou et al. (2020) [3]. In these two papers, convergence rates are obtained in the SBM for the parameters $\alpha$ and $\mathbf{Q}$. By contrast, we focus on estimating the matrix of connection probabilities $\mathbf{Theta}$, which requires estimating the community assignments. Thus, our results do not follow from that of Depdeep et al. (2018) [1], in the same way that they do not follow from that of Bickel. et al. (2013) [2] and Mariadassou et al. (2020) [3]. On the other hand, it would be interesting to investigate whether the techniques of proof we use to study the label assignments can be applied in the more general framework described in Depdeep et al. (2018) [1].
>
> Finally, we also underline that the methods and results presented in Depdeep et al. (2018) [1], are obtained for an estimator that is different to that studied in Bickel. et al. (2013) [2] and Mariadassou et al. (2020) [3](because of the introduction of the prior distribution on the parameters).
>
> In Yang et al. (2020)[4], the authors extend their results to more general variational objective functions, and more general (unbounded) parameter spaces. The main differences in our model and results remain however the same.
>
> We also thank the reviewers for their suggestions on how to improve the clarity of our paper, and for pointing out remaining typos in the manuscript.
>
> [1] Pati, Debdeep, Anirban Bhattacharya, and Yun Yang. "On statistical optimality of variational Bayes." International Conference on Artificial Intelligence and Statistics. PMLR, 2018.
> [2]Peter Bickel, David Choi, Xiangyu Chang, and Hai Zhang. "Asymptotic normality of maximum likelihood and its variational approximation for stochastic blockmodels". The Annals of Statistics, 41(4):1922 – 1943, 2013.
> [3]Mahendra Mariadassou and Timothee Tabouy." Consistency and asymptotic normality of stochastic block models estimators from sampled data". Electronic Journal of Statistics, 14(2):3672 – 3704, 2020.
> [4] Yang, Yun, Debdeep Pati, and Anirban Bhattacharya. "α-variational inference with statistical guarantees." The Annals of Statistics 48.2 (2020): 886-905

---

### Decision · Program_Chairs · 2021-09-27

**Decision:**

Accept (Poster)

**Comment:**

The paper proves that the mean-field variational inference is minimax optimal for estimating the parameters of the stochastic block model (SBM) even when some of the links are missing. While the mean-field variational method has been studied for the SBM, this paper still addresses an interesting question. In particular, it shows that there is no computational gap in the problem of link prediction for SBM byproviding an estimator that is minimax optimal and computationally feasible (this question had been open for a while). Hence, given this contribution, I agree with the reviewers that the paper should be accepted in Neurips.